# Postural and Muscular Responses to a Novel Multisensory Relaxation System in Children with Autism Spectrum Disorder: A Pilot Feasibility Study

**DOI:** 10.3390/children12111455

**Published:** 2025-10-26

**Authors:** Laura Zaliene, Daiva Mockeviciene, Eugenijus Macerauskas, Vytautas Zalys, Migle Dovydaitiene

**Affiliations:** 1Department of Holistic Medicine and Rehabilitation, Faculty of Health Science, Klaipeda University, LT-92294 Klaipėda, Lithuania; daiva.mockeviciene@ku.lt; 2Department of Electrical and Electronic Engineering, Faculty of Technic, Vilniaus Kolegija/Higher Education Institution, LT-08106 Vilnius, Lithuania; e.macerauskas@tef.viko.lt; 3Department of Electrical Engineering, Faculty of Electronics, Vilnius Gediminas Technical University, LT-10223 Vilnius, Lithuania; 4Department of Engineering and Technology, Faculty of Bossiness end Technology, Utenos Kolegija/High Education institution, LT-28176 Utena, Lithuania; 5Siauliai Academy, Vilnius University, LT-01513 Vilnius, Lithuania; zalysvytautas@yahoo.com; 6Institute Integrative and Body Psychotherapy, Vilnius University, LT-03224 Vilnius, Lithuania; institutas@ikpi.lt

**Keywords:** autism spectrum disorder, posture, electromyography, muscle activity, sensory integration, relaxation system

## Abstract

**Highlights:**

**What are the main findings?**
A smart relaxation system was safe and well tolerated by children with severe autism spectrum disorder.No significant increases in muscle tension were detected; physiological relaxation effects were observed.

**What is the implication of the main finding?**
The system shows promise as a short-term calming intervention for children with autism.Findings provide a foundation for developing multisensory approaches in education and rehabilitation.

**Abstract:**

Background: Children with autism spectrum disorder (ASD) frequently show postural abnormalities and elevated muscle tone, which can hinder participation in education and rehabilitation. Evidence on the immediate physiological effects of standardized multisensory environments is limited. Objective: To evaluate feasibility, safety and short-term physiological/postural responses to an automated multisensory smart relaxation system in children with severe ASD. Methods: In a single-session pilot across three sites, 30 children (27 boys; 6–16 years) underwent pre–post postural observation and bilateral surface EMG of the upper trapezius, biceps brachii and rectus abdominis. The system delivered parameterized sound, vibration, and mild heat. EMG was normalized to a quiet-sitting baseline. Results: The intervention was well tolerated with no adverse events. Most children sat independently (25/30; 80%) and a majority stood up unaided after the session (24/30; 76.9%). Postural profiles reflected common ASD features (neutral trunk 76%, forward head 52%, rounded/protracted shoulders 46%), while limb behavior was predominantly calm (73%). Normalized EMG amplitudes were low, with no significant pre–post changes and no meaningful left–right asymmetries (all *p* > 0.05; Cohen’s *d* < 0.20), indicating physiological calmness rather than tonic co-contraction. Conclusions: A single session with a smart multisensory relaxation system was safe, feasible, and physiologically calming for children with severe ASD, without increasing postural or muscular tension. The platform’s standardization and objective monitoring support its potential as a short-term calming adjunct before therapy or classroom tasks. Larger, gender-balanced, multi-session trials with behavioral outcomes are warranted.

## 1. Introduction

Autism spectrum disorder (ASD) affects about 1 in 100 children worldwide and is characterized not only by social–communication difficulties but also by atypical responses to sensory input [1]. Many children show motor and postural challenges—forward-head/rounded-shoulder posture, dyspraxia, clumsiness, toe-walking, joint hypermobility, and balance problems—which can limit participation in everyday activities and education [2,3].

Sensory-based approaches are widely used to support self-regulation and functional engagement in ASD [4,5,6]. However, commonly used options have limitations: Snoezelen-type rooms provide soothing input but are difficult to standardize and quantify; massage-based interventions may reduce arousal but show mixed evidence and provider-dependent variability [4,5,6]. Children with ASD also frequently present heightened emotional reactivity and self-regulation difficulties that burden families and hinder inclusion in education and rehabilitation [7].

Technology-assisted tools (e.g., music/sound systems and virtual reality) can engage attention but may provoke overload in some users and often lack objective physiological endpoints [8,9]. Objective physiological endpoints in multisensory contexts are limited and often focus on measures such as hear rate variability (HRV) during sleep [10], with little attention to neuromuscular tone (surface electromyography-sEMG) and posture. Against this backdrop, we tested an automated, parameterized multisensory system that delivers reproducible sound, vibration, and mild thermal stimuli and incorporates objective monitoring (surface electromyography and structured postural observation). The platform is designed for standardization, dose control, and scalability within school and rehabilitation settings.

We used a single-session, pre–post design to answer early-phase questions of feasibility, safety, and immediate physiological effect. A brief exposure minimizes burden for children with severe ASD, aligns with pilot feasibility methodology, and isolates short-term neuromuscular responses that can inform dosing and the design of future multi-session trials. We hypothesized that a single, standardized session would be well tolerated and associated with physiological calmness—operationalized as low, symmetrical sEMG activity without worsening of postural features—supporting the system’s potential as a short-term calming adjunct before therapy or classroom tasks.

## 2. Materials and Methods

### 2.1. Participants and Setting

The study was conducted at three sites: Klaipėda University’s Department of Holistic Medicine and Rehabilitation, Ringuva School (Siauliai), and Vilnius State College. Thirty children with severe autism spectrum disorder (ASD) participated (27 boys, 3 girls; age 6–16 years; mean 10.3 ± 2.5). Given the small and gender-imbalanced sample, sex-stratified analyses were not planned.

### 2.2. Study Design and Rationale

We used a single-group, pre–post feasibility design to address early-phase questions of feasibility, safety, and immediate physiological effects in children with severe ASD. A randomized or sham-controlled design was not pursued at this stage to (i) minimize burden and reduce the risk of sensory overload from repeated or deceptive exposures, (ii) accommodate pragmatic constraints in school/rehabilitation timetables (a single 15–30 min session per child across three sites), and (iii) first characterize within-participant physiological change given high inter-individual heterogeneity. Pre- and post-assessments occurred within minutes in the same session, reducing history/maturation threats. Outcomes were objective (surface electromyography and structured postural observation). We acknowledge that this design limits interpretability and does not permit causal inference; findings provide feasibility metrics and effect-size estimates to inform subsequent controlled, multi-session trials (see Limitations).

### 2.3. Eligibility Criteria

Inclusion: (1) Documented DSM-5 diagnosis of severe ASD by a licensed clinician; (2) age 6–16 years; (3) no contraindications to using the Smart Relaxation System (SRS); (4) stable psychoactive medication for ≥4 weeks prior to the session; (5) no concurrent physiotherapy/occupational therapy or experimental sensory interventions during the study and for ≥2–4 weeks prior; (6) written informed consent from legal guardians. Exclusion: (1) Sensory phobias that would preclude safe exposure (e.g., intense aversion to color, sound, or vibration); (2) profound intellectual disability precluding assent or safe participation; (3) acute illness/infection; (4) chronic conditions making participation unsafe; (5) known neuromuscular or orthopedic disorders materially affecting EMG/posture (e.g., recent casting or orthopedic surgery, botulinum toxin within 6 months).

### 2.4. Recruitment and Sample Flow

Recruitment took place from February to May 2023 across three sites (Klaipėda University’s Department of Holistic Medicine and Rehabilitation; Ringuva School, Siauliai; Vilnius State College). Families responded to an open call posted by the Lietaus Vaikai Association and were screened by the study team for eligibility. Thirty children who met inclusion criteria were scheduled for one individual SRS session. Study sessions were conducted from June to August 2023. All 30 participants completed the protocol; there was no loss to follow-up due to the single-session design.

### 2.5. Procedures

Electrode placement sites were identified by palpation according to SENIAM guidelines. Six bipolar surface electrodes (Biometrics Ltd., Newport, UK) were placed bilaterally over the upper trapezius, biceps brachii, and rectus abdominis; inter-electrode distance 20 mm; reference electrode over the ulnar styloid. Skin was shaved and cleaned with alcohol to reduce impedance. EMG was acquired using a 6-channel system at 5000 Hz; signals were band-pass filtered 20–450 Hz, notch-filtered at 50 Hz, rectified, and RMS-processed (20 ms window). Normalization used each participant’s 10 s quiet-sitting baseline immediately prior to the session. A standardized maximal voluntary contraction (MVC) protocol was not feasible in this severe ASD cohort due to compliance constraints; consequently, results reflect within-participant relative change only. The absence of MVC normalization is a methodological limitation and is addressed in the Limitations section. During the session, participants sat in the SRS chair and were instructed to relax; no cognitive or motor tasks were imposed, and no restraints were used. Postural behavior was assessed with a structured observational protocol (Table 1). For each participant, the most frequently observed posture during the session was recorded.

### 2.6. Outcomes

Primary feasibility/safety endpoints were tolerability, adverse events, and session completion. Physiological endpoints included bilateral surface EMG from the upper trapezius, biceps brachii, and rectus abdominis, and postural behavior categories (head, trunk, shoulders, upper/lower limbs).

### 2.7. Handling of Missing and Artifact Data

EMG channels with obvious motion or electrical artifact were excluded after independent visual review by two assessors; discrepancies were resolved by consensus. For left–right symmetry analyses, if one channel of a bilateral muscle was excluded for a participant, the paired channel for that muscle was also excluded to avoid biased comparisons. Per-analysis denominators (n) are reported in the relevant tables and figure captions. Postural categories with incomplete observation were coded as missing and excluded from category-specific percentages.

### 2.8. Smart Relaxation System (SRS)

The SRS is a prototype sensory-integrative environment designed to promote physiological relaxation. The system consists of:A massage chair with integrated vibratory and tactile transducers (seat and lateral components);An embedded sound system with bass and surround speakers;color light sources mounted in a movable hood structure;A video camera for facial monitoring linked to AI-driven scenario adjustments;A user interface controlled via a touch-screen computer (Figure 1, Figure 2, Figure 3 and Figure 4).

Auditory stimuli were delivered at 50–60 dB SPL with frequencies between 250 and 2000 Hz. Vibration was applied at 30–45 Hz with an amplitude of 0.3–0.5 mm. A thermal element maintained a steady surface temperature of ~38 °C.

Each session lasted 15–30 min, depending on individual tolerance and behavioral readiness. After the intervention, surface EMG and postural observation data were processed and analyzed.

### 2.9. Statistical Analysis

Analyses were conducted in IBM SPSS v24.0 (Excel was used only for figure preparation). Normality was assessed with the Shapiro–Wilk test. For EMG, we analyzed within-participant left–right differences for each muscle using paired-sample t-tests (or Wilcoxon signed-rank when normality was violated). Between-muscle contrasts were summarized descriptively, because signals were normalized to a resting baseline (no MVC) and muscles have distinct functions. Effect sizes for paired comparisons are reported as Cohen’s dz (mean of paired differences/SD of paired differences) with conventional thresholds (0.2 small, 0.5 medium, 0.8 large). Two-tailed α = 0.05. Given the exploratory feasibility scope and the small sample, we emphasized estimation over confirmatory inference and did not apply multiplicity adjustments. Per-analysis denominators (n) are provided in the Results/figure captions.

### 2.10. Ethics

The study complied with the Convention on Human Rights and Biomedicine (adopted 19 November 1996). Legal guardians provided written informed consent. Ethics approval: Klaipėda University Bioethics Committee (HMRK-BE-22-15).

## 3. Results

### 3.1. Postural Behavior

The results of body behavior assessment are summarized in Table 2. Postural-behavior assessment indicated good feasibility and tolerability. Most children sat independently in the SRS chair (25/30; 80.0%), with some seated by a caregiver (3/30; 12.0%). At the end of the session, the majority also stood up unaided (24/30; 76.9%), while a few required assistance (5/30; 19.2%). Neutral trunk alignment predominated (20/30; 76.0%), with a hunched thoracic posture observed less frequently (6/30; 23.0%). Forward-head posture was common (13/30; 52.0%), although a subset maintained a neutral head position (9/30; 36.0%). Shoulder alignment was neutral in half of the sample (13/30; 50.0%), and rounded/protracted in nearly as many (12/30; 46.0%). The upper and lower limbs were most often at rest (each 19/30; 73.1%), while stereotyped movements were noted in 6/30 (23.1%). This profile—predominantly neutral trunk posture, calm limb behavior, and the typical combination of forward head and shoulder protraction—accords with features described in ASD and, together with low, symmetrical sEMG amplitudes, indicates that the intervention did not increase postural or muscular tension and that the environment was safe and practically applicable in educational and rehabilitation settings. We did not observe behavioral signs of increased postural strain during exposure.

### 3.2. Electromyographic (EMG) Activity

Surface EMG was recorded bilaterally from the upper trapezius (UT), biceps brachii (BB), and rectus abdominis (RA), normalized to baseline levels (Figure 5).

Surface EMG was recorded continuously during the session and normalized to a quiet-sitting baseline. Analyses focused on left–right symmetry within each muscle. For the upper trapezius (UT), mean normalized RMS was 0.075 ± 0.188 (left) vs. 0.040 ± 0.077 (right); the paired mean difference was 0.035 (SD of differences = 0.177), dz = 0.20, *p* = 0.439 (*n* = 16). For biceps brachii (BB), values were 0.012 ± 0.014 (left) vs. 0.013 ± 0.018 (right); difference −0.001 (SD = 0.006), dz = −0.15, *p* = 0.565 (*n* = 15). For rectus abdominis (RA), values were 0.024 ± 0.049 (left) vs. 0.017 ± 0.032 (right); difference 0.006 (SD = 0.050), dz = 0.12, *p* = 0.640 (*n* = 15). All side differences were non-significant and effect sizes trivial (dz < 0.20), indicating no meaningful left–right asymmetry during exposure. Group means were uniformly low in amplitude, compatible with relaxed tonic activation; however, as an observational feasibility dataset (no pre–post EMG), these values should not be interpreted as evidence of physiological change. Taken together, these results show that the Smart Relaxation System did not increase muscle tension or postural strain. On the contrary, consistently low EMG activity across trunk and shoulder muscles suggests a state of reduced arousal and possible physiological calmness during the intervention. No adverse events occurred; sessions were feasible in school/rehab settings.

### 3.3. Subgroup Analyses

Given the small and strongly gender-imbalanced sample (27 boys, 3 girls) and the single-session feasibility scope, pre-specified subgroup analyses (e.g., by sex or severity) were not performed. Such analyses would be underpowered and potentially misleading. We highlight this as a limitation and plan sex-balanced recruitment and stratified analyses in subsequent controlled studies.

## 4. Discussion

The aim of this study was to examine the effects of a multisensory smart relaxation system on postural behavior and muscle activity in children with autism spectrum disorder (ASD). The system integrates auditory, vibratory, thermal, and visual stimuli designed to deliver standardized multisensory input within a controlled environment. We did not assess sensorimotor regulation per se.

### 4.1. Safety and Feasibility

Clinically, the intervention was safe, feasible, and well tolerated. No adverse events occurred, and most children engaged with the environment without distress. Postural deviations such as hunched back, forward head posture, and stereotyped movements were consistent with known motor features of ASD [11,12], rather than induced by the intervention. This pattern is consistent with links between motor coordination and social–emotional behaviour in early childhood (see also Piek et al., 2008 [13]). Single-case and small-series reports suggest that controlled vibrotactile input can reduce motor stereotypies in some children (e.g., whole-body vibration; Bressel et al., 2011 [14]), but these observations require replication in larger, controlled samples. Similar challenges have been reported in developmental coordination disorder (DCD), ADHD, and intellectual disability [15,16], suggesting potential applicability beyond ASD pending controlled evaluation.

Practical/clinical link: Because sessions were well tolerated and did not increase apparent postural or muscular load, this brief, standardized exposure can be considered as a preparatory adjunct before physiotherapy/occupational therapy tasks or classroom transitions. Such a predictable, low-burden routine may help teams establish entry routines; however, we make no claims about efficacy and will test these functional/behavioral effects in controlled, multi-session trials.

### 4.2. Muscle Activity

Electromyographic recordings showed low, symmetrical activation across the upper trapezius, biceps brachii, and rectus abdominis. No significant left–right differences were observed, and Cohen’s *d* values were trivial (<0.20). These results indicate no added tonic co-contraction during exposure. Because autonomic indices were not collected and MVC normalization was not applied, these EMG observations should not be interpreted as evidence of physiological change. Accordingly, we refrain from terms such as ‘physiological calmness’ and do not infer autonomic change from EMG alone. Clinically, this reassures practitioners when planning tasks that benefit from relative muscular relaxation (e.g., fine-motor/handwriting training, feeding skills, balance tasks).

### 4.3. Mechanistic Perspective

We deliberately refrain from mechanistic inference in this study. While multisensory stimulation can engage cortical and subcortical networks implicated in movement, attention, and emotion regulation, and rhythmic auditory/vibratory input has been associated with effects on sensorimotor integration and stress modulation [17,18,19]. However, we did not collect autonomic or neurophysiological markers in this study, so any mechanistic explanation (e.g., autonomic regulation or cortical engagement) remains hypothetical and should be tested in future work using HRV/EDA, pupillometry, EEG/fNIRS, alongside clearly defined clinical/behavioral endpoints [20]. In addition, atypical processing of affective touch in autism provides a plausible sensory pathway for future study [21].

### 4.4. Comparison with Existing Approaches

Our results conceptually align with multisensory environments such as the “Magic Room,” which have reported improvements in engagement and self-regulation in children with neurodevelopmental conditions [21]. The smart relaxation system extends this concept by integrating physiological monitoring, thereby offering both therapeutic and assessment functions. Compared with existing options, Snoezelen-type spaces can be soothing, yet are difficult to standardize or quantify across sessions and sites; massage may reduce arousal in some children but the evidence is mixed and effects are provider-dependent [4,5,6]; and music/sound or VR tools can enhance engagement but may precipitate sensory overload in sensitive users and often lack objective neuromuscular endpoints [8,9]. Consistent with this, a systematic review of sensory-based interventions for behavioural issues in children reported heterogeneous methods and generally limited/low-to-moderate quality evidence, with cautious conclusions about efficacy [22]. In contrast, the present platform delivers parameterized, dose-controlled sound/vibration/heat with objective monitoring (structured posture observation and sEMG), enabling reproducibility and scalability across educational and rehabilitation settings. This complements prior ‘smart space’ concepts [14] and provides a framework to add a measurable physiological layer in future studies that include autonomic/neural outcomes.

## 5. Limitations

This pilot has several important limitations. First, the single-group, single-session design without a control condition precludes causal inference and limits interpretability to feasibility/safety signals. Second, the sample was small and strongly male-predominant (27 boys, 3 girls), which restricts generalizability and prevented stratified analyses (e.g., by sex or ASD severity). Third, surface EMG was normalized to a quiet-sitting baseline rather than MVC; while appropriate for this cohort, it limits comparability across participants and muscles and precludes conclusions about absolute muscle load. Fourth, we did not collect behavioral or psychological outcomes (e.g., on-task behavior, affect regulation, caregiver/teacher ratings), so functional implications cannot be inferred. Fifth, autonomic or neurophysiological markers (e.g., HRV/EDA, EEG/fNIRS) were not recorded; therefore, mechanistic interpretations remain hypothetical. Finally, posture ratings were conducted with a structured protocol across three sites, but we did not quantify inter-rater reliability, and brief single-session data cannot speak to durability of effects.

## 6. Clinical Implications and Future Research

Within these constraints, the data support feasibility and safety of a brief, standardized, multisensory exposure that did not increase observable postural strain or tonic EMG activation during the session. Clinically, such a predictable, low-burden routine could be explored as a preparatory adjunct before therapy tasks or classroom transitions; however, any functional/behavioral benefits must be demonstrated in controlled studies. Future work should employ controlled, multi-session designs with sex-balanced, adequately powered samples; pre-register primary outcomes and statistical plans; include validated behavioral/psychological measures (e.g., participation/on-task metrics, caregiver/teacher ratings, adaptive behavior, sensory reactivity); and add acceptability/usability and qualitative feedback from caregivers and teachers to enhance ecological validity. Mechanistic testing should incorporate autonomic indices (HRV/EDA), pupillometry, and portable EEG/fNIRS. For EMG, consider submaximal reference tasks or RVE-type normalization when MVC is not feasible, and report inter-rater reliability for posture observation. Objective kinematic measures (e.g., IMUs) and longer follow-up would help link session responses to real-world participation.

## 7. Conclusions

In this multi-site pilot with children who have severe ASD, a single standardized session of an automated multisensory relaxation system was feasible and well tolerated, with no adverse events. Observational posture data and normalized surface EMG recorded during the session did not indicate added postural load or tonic co-contraction, and left–right differences were not statistically significant. These findings should be interpreted strictly as feasibility/safety signals; the single-group, single-session design does not permit conclusions about efficacy or mechanisms. Controlled, multi-session studies with larger, gender-balanced samples and validated behavioral, autonomic, and neurophysiological outcomes are required to determine clinical utility and mechanism of action. The platform’s parameterization and objective monitoring may facilitate standardized implementation and hypothesis testing in future trials.

## Figures and Tables

**Figure 1 children-12-01455-f001:**
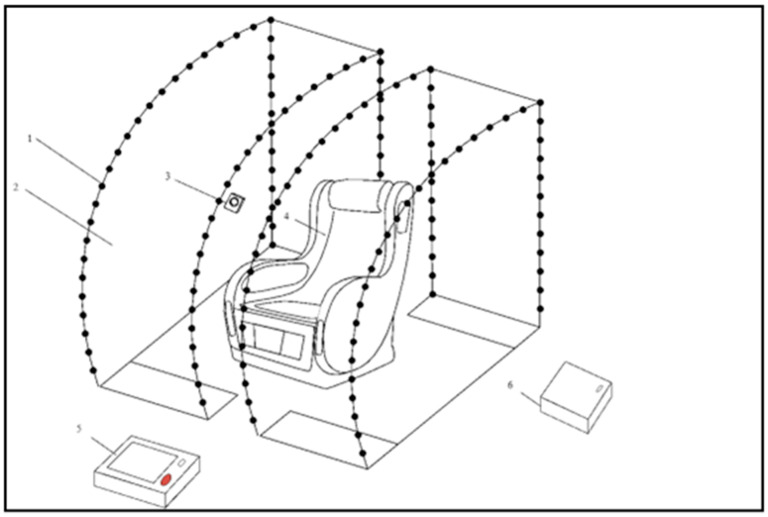
System components: 1—semi-transparent hood (left and right parts), 2—system of colored light sources, 3—video camera, 4—massage chair with sound system, 5—control computer with touch screen, 6—external power source.

**Figure 2 children-12-01455-f002:**
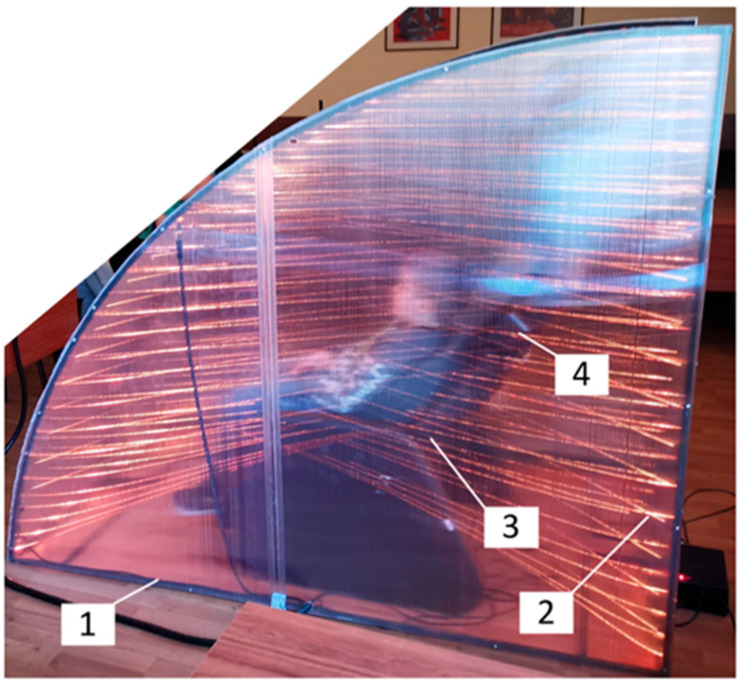
Real image of the prototype together with the child subject (side projection): 1—semi-transparent cover, 2—light system, 3—relaxation chair, 4—sound system (rear part).

**Figure 3 children-12-01455-f003:**
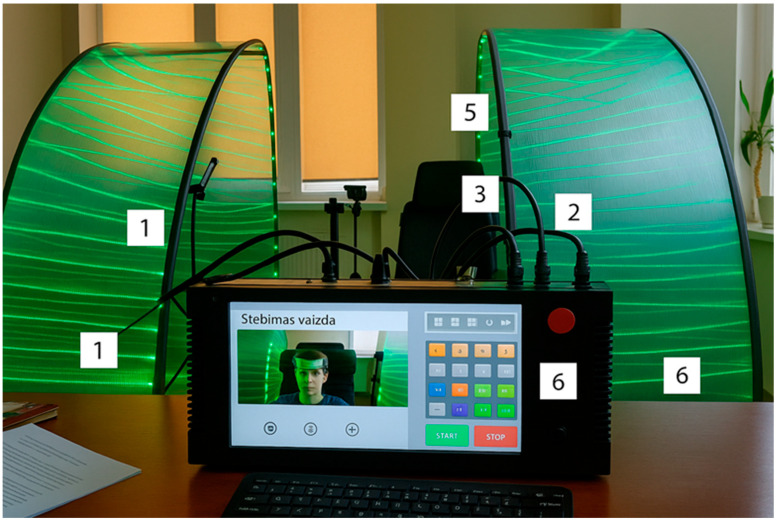
Real image of the prototype together with the child subject (front projection): 1—semitransparent cover (left and right parts), 2—light system, 3—relaxation chair, 4—sound system (rear part), 5—video camera, 6—control computer.

**Figure 4 children-12-01455-f004:**
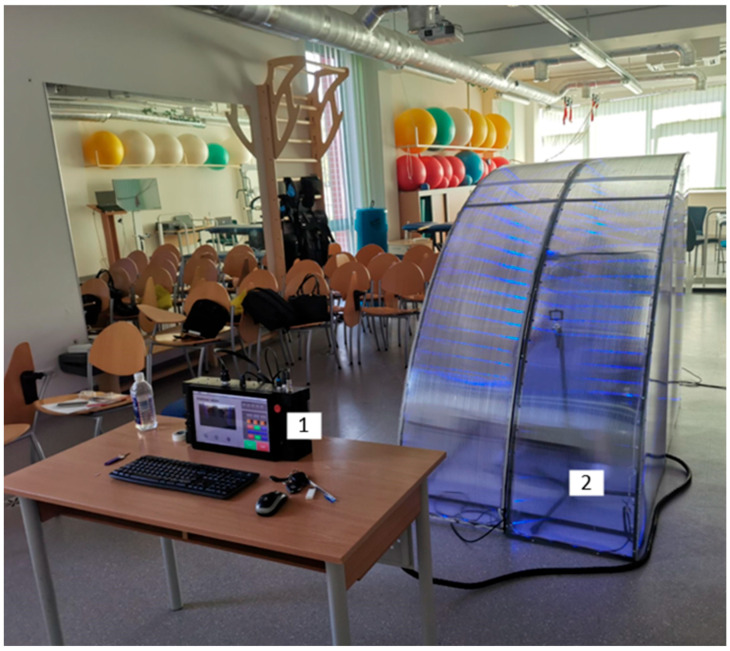
An overview of the entire system in real operating conditions: 1—control computer, 2—relaxation system with a chair inside.

**Figure 5 children-12-01455-f005:**
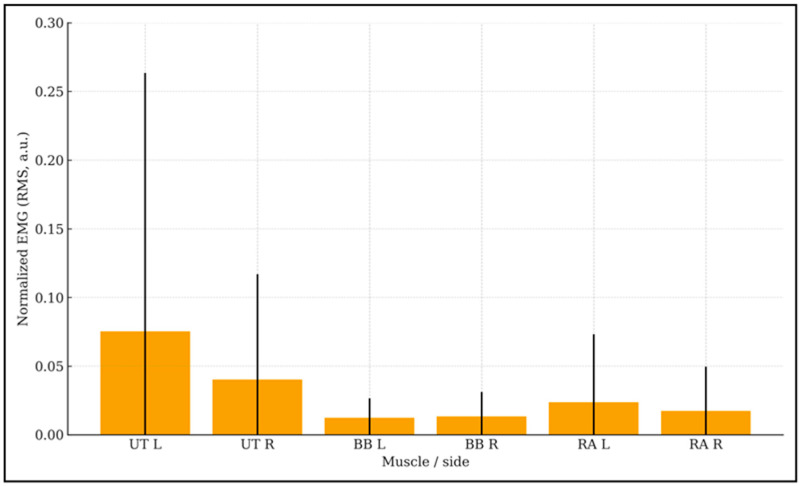
Normalized surface EMG (RMS, a.u.) during the session by muscle and side (mean ± SD). Bars are normalized to a 10 s quiet-sitting baseline per participant. Sample sizes (left/right): UT 30/30, BB 30/30, RA 30/30. Abbreviations: UT—upper trapezius; BB—biceps brachii; RA—rectus abdominis; L/R—left/right. No significant left–right differences; all Cohen’s dz < 0.20.

**Table 1 children-12-01455-t001:** Categories of Body Behavior Observed (operational definitions and coding).

Body Behavior	Fixing Categories	Description	Coding (0/1)	Notes/Examples
Seating (entry)	Placed by caregiver	Child is seated in the SRS chair by a caregiver/therapist.	0 = absent; 1 = present	Former “Being planted”.
	Sits with assistance	Child steps up and sits with physical guidance.	0/1	Hand-over-hand, trunk guidance.
	Sits independently	Child sits from step or floor without assistance.	0/1	Feasibility indicator.
Trunk position	Neutral	Head and spine aligned; pelvis neutral for ≥50% of the session (cumulative).	0/1	
	Forward lean	Trunk flexed forward and/or anterior pelvic tilt for ≥50% of the session.	0/1	
	Backward lean	Trunk extension and/or posterior pelvic tilt for ≥50% of the session.	0/1	
	Hunched posture	Excess thoracic kyphosis beyond neutral for ≥50% of the session.	0/1	Record co-occurrence with forward head.
Head position	Neutral	Head in neutral alignment; supported by chair back as needed, ≥50% of the session.	0/1	
	Forward head	Anterior translation of head relative to trunk in sitting for ≥50% of session	0/1	Code protraction rather than simple neck flexion.
	Backward head	Posterior translation of head relative to trunk in sitting for ≥50% of session	0/1	
	Asymmetrical	Sustained lateral flexion and/or rotation (left/right) for ≥50% of the session.	0/1	
Shoulder	Neutral	No protraction/elevation; symmetrical scapular position for ≥50% of the session.	0/1	
	Leaning forward	Visible anterior rounding or elevation of shoulders for ≥50% of session	0/1	Differentiate postural habit vs. transient movement.
	Asymmetrical	One shoulder elevated/depressed relative to the other for ≥50% of the session.	0/1	
Arm position	At rest	Hands on armrests or next to torso; minimal movement; no tonic co-contraction for ≥50% of the session.	0/1	Use alongside EMG interpretation.
	Stereotyped movements	Repetitive, non-functional limb movements (e.g., flapping) occurring ≥3 bouts	0/1	Record frequency if feasible.
	Tense	Sustained visible co-contraction/rigidity ≥10 s and/or resistance to passive repositioning.	0/1	
Legs	At rest	Legs supported on chair; minimal movement; no tonic co-contraction for ≥50% of the session.	0/1	Use alongside EMG interpretation.
	Stereotyped movements	Repetitive, non-functional limb movements (e.g., flapping) occurring ≥3 bouts	0/1	Record frequency if feasible.
	Tense	Sustained visible co-contraction/rigidity ≥10 s and/or resistance to passive repositioning.	0/1	
Exit	Removed by caregiver	Child is lifted/guided out of the chair by caregiver/therapist.	0/1	
	Exits with assistance	Child stands and steps down with guidance.	0/1	
	Exit Independently	Child stands and steps down without assistance.	0/1	Feasibility/tolerability indicator.

Notes: “≤50% of the session” denotes cumulative duration over the observation period. Multiple categories may co-occur and should each be coded as applicable. Raters were trained on the protocol and discrepancies resolved by consensus. Abbreviations: SRS, Smart Relaxation System; EMG, electromyography.

**Table 2 children-12-01455-t002:** Key Body Behavior Results (N = 30).

Category	Observed Behavior	Frequency (N)	Percentage (%)
Getting off the chair	Independently	24	76.9
Getting off the chair	With assistance	5	19.2
Head posture	Forward head	13	52.0
Head posture	Correct	9	36.0
Lower extremities	Calm	19	73.1
Lower extremities	Stereotyped movements	6	23.1
Shoulder	Neutral	13	50.0
Shoulder	Rounded/protracted	12	46.0
Sitting in a chair	Independently	25	80.0
Sitting in a chair	Placed by caregiver	3	12.0
Trunk posture	Neutral	20	76.0
Trunk posture	Hunched posture	6	23.0
Upper extremities	At rest (calm)	19	73.1
Upper extremities	Stereotyped movements	6	23.1

Notes. Percentages are based on the total sample (n = 30) and rounded to one decimal place. Within domain totals may not equal 100% because additional categories (e.g., “with assistance” for seating, “asymmetrical” head/shoulder posture, or “tense” limbs) are not displayed here. Terminology harmonized with Table 1: “Forward head” (formerly “bended forward”), “Rounded/protracted shoulders” (formerly “leaning forward”), and “Placed by caregiver” (formerly “planted”).

## Data Availability

The data presented in this study are available on request from the corresponding author. The data are not publicly available due to ethical restrictions related to the protection of participants’ privacy.

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
