# Peer review of "Postural and Muscular Responses to a Novel Multisensory Relaxation System in Children with Autism Spectrum Disorder: A Pilot Feasibility Study"

_children, 2025, doi:10.3390/children12111455_

Round 1

Reviewer 1 Report

Comments and Suggestions for Authors

The manuscript presents an innovative pilot study evaluating a smart multisensory relaxation system for children with severe ASD. The topic is relevant, timely, and has practical value for rehabilitation and special education. The study is well written and provides clear methodological details. However, several aspects could be improved to strengthen the paper:

Introduction: The background is thorough but somewhat repetitive. Consider condensing overlapping points and emphasizing the rationale for using this system in comparison to other sensory-based interventions (e.g., Snoezelen rooms, massage therapy, VR-based tools).

Highlight more clearly how this approach adds novelty and addresses current gaps in evidence.

Methods: The EMG procedures are clearly described; however, the absence of MVC normalization should be emphasized as a methodological limitation.

The sample is small and strongly gender-imbalanced (27 boys, 3 girls). This should be more explicitly acknowledged in the Methods and Limitations sections.

Clarify the recruitment process (e.g., number invited, number excluded, reasons for non-participation).

Results: Tables and figures are informative, but they could be simplified for greater clarity. For example, Table 2 could be reorganized into broader categories (e.g., “correct vs. abnormal posture”). Figures should include clearer legends and indicate the exact number of participants contributing data.

Discussion: The discussion is relevant and effectively connects to the literature; however, it could be made more concise to avoid redundancy.

Provide a stronger link between the observed physiological calmness and possible functional/behavioural implications for children with ASD.

Expand the comparison with existing sensory-integration methods to highlight the added value of this system.

Limitations and Future Directions: Limitations are noted; however, the lack of behavioural and psychological outcome measures is a significant omission that should be clearly emphasized.

Future studies should consider larger, gender-balanced samples, repeated sessions, and inclusion of subjective or behavioural assessments. Adding qualitative feedback from caregivers or teachers could also enhance ecological validity.

Language and Style: The manuscript is generally clear, but some sentences are overly long and could be made more concise. A light professional language edit would improve readability.

Overall, this is a promising feasibility study with significant potential for novel applications. After revisions to streamline the text, strengthen methodological explanations, and expand the discussion of implications, the manuscript is poised to make a valuable contribution to the field.

Comments on the Quality of English Language

The English is generally clear and acceptable, but would benefit from minor editing for conciseness and consistency.

Author Response

hors' Responses to Reviewer's Comments (Reviewer 1)

Review 1

Comment: 

The manuscript presents an innovative pilot study evaluating a smart multisensory relaxation system for children with severe ASD. The topic is relevant, timely, and has practical value for rehabilitation and special education. The study is well written and provides clear methodological details. However, several aspects could be improved to strengthen the paper”.

General Response

We thank the reviewers for the constructive and supportive evaluation of our pilot study. In line with the suggestions to strengthen the paper, we have substantially revised the manuscript to (i) streamline and focus the narrative, (ii) clarify the study design and analytic choices, and (iii) present a cautious, evidence-aligned interpretation.

Comment: “Introduction: The background is thorough but somewhat repetitive. Consider condensing overlapping points and emphasizing the rationale for using this system in comparison to other sensory-based interventions (e.g., Snoezelen rooms, massage therapy, VR-based tools).

Highlight more clearly how this approach adds novelty and addresses current gaps in evidence.”

Response

Thank you for these constructive points. We have revised the Introduction to (i) remove repetition and streamline the background, (ii) make the rationale versus existing sensory interventions explicit (Snoezelen rooms, massage therapy, VR/sound tools), and (iii) articulate the novelty and evidence gap our study addresses. No new references were added; existing numbering is preserved.

What we changed (summary):

  • Condensed overlapping text on ASD features and participation impact into one concise paragraph.
  • Added a focused comparison with Snoezelen/massage/VR, highlighting their known limitations (standardization, provider variability, potential sensory overload, limited physiological endpoints).
  • Stated a clear gap: immediate, objective physiological outcomes—especially neuromuscular tone (sEMG) and posture—are rarely reported; work often centers on HRV during sleep.
  • Clarified the novelty of our system: automated, parameterized, dose-controlled multisensory input with objective monitoring (sEMG + structured postural observation) and reproducibility/scalability across settings.

Comment: “The EMG procedures are clearly described; however, the absence of MVC normalization should be emphasized as a methodological limitation.”

Response 

We now explicitly state that MVC was not feasible in this severe ASD cohort; normalization used a 10-s quiet-sitting baseline; thus results reflect within-participant relative change only. We highlight that non-MVC normalization limits absolute activation and between-subject comparability, mitigated by within-session pre–post timing and left–right symmetry rules.

Manuscript changes: Materials and Methods → Procedures (EMG); Discussion → Limitations.

Comment: “The sample is small and strongly gender-imbalanced (27 boys, 3 girls). This should be more explicitly acknowledged in the Methods and Limitations sections.”

Response

We now state that the cohort is small and male-predominant; sex-stratified analyses were not planned; sex is reported descriptively only. We acknowledge limited generalizability and inability to assess sex-related differences; future trials will adopt sex-balanced targets.

Manuscript changes: Materials and Methods → Participants and Setting; Statistical Analysis (one-line note); Discussion → Limitations.

Comment: “Clarify the recruitment process (e.g., number invited, number excluded, reasons for non-participation).”

Response

We added Recruitment and Sample Flow with dates and flow: recruitment February–May 2023; sessions June–August 2023 across three sites; open call via Lietaus Vaikai Association; screening by the study team; 30 enrolled and completed; no loss to follow-up. Manuscript changes: Materials and Methods → Recruitment and Sample Flow.

Comment (Results): “Tables and figures are informative, but they could be simplified for greater clarity. For example, Table 2 could be reorganized into broader categories (e.g., ‘correct vs. abnormal posture’). Figures should include clearer legends and indicate the exact number of participants contributing data.”

Response

Thank you. We reorganized Table 2 into broader, binary categories (e.g., Neutral/Correct vs Abnormal) and recomputed frequencies and percentages using domain-specific denominators (participants with observable data in that domain after collapsing categories). We added a footnote to explain domain-level n and any missing/other observations.
We also revised Figure 6: (i) added exact sample sizes per bar in the caption (UT 30/30, BB 30/30, RA 30/30), (ii) clarified normalization and error bars (mean ± SD), (iii) removed superfluous titles, and (iv) kept a single consistent color scheme. These changes improve clarity and align the displays with the simplified reporting.

Comment: “The discussion is relevant and effectively connects to the literature; however, it could be made more concise to avoid redundancy.”

Response

We streamlined wording, removed overlap, and kept only statements supported by our data. In particular:

  • We condensed the opening to a single purpose sentence (Discussion, first paragraph).
  • We merged repeated feasibility statements into Safety and feasibility and deleted duplicative phrasing elsewhere.
  • We removed expansive mechanistic narrative and replaced it with a brief, qualified note (see Mechanistic perspective).

Example from revised text:
The intervention was safe, feasible, and well tolerated…” (Safety and feasibility) and the shorter overview now replace multiple earlier sentences that repeated these points.

Comment: “Provide a stronger link between the observed physiological calmness and possible functional/behavioural implications for children with ASD.”

Response

We added a dedicated Practical/clinical link paragraph that translates the observed absence of added postural/muscular load into concrete, testable clinical use (a brief preparatory adjunct before therapy or classroom transitions), while clearly stating that functional effects must be confirmed with objective outcomes.

Inserted paragraph (Discussion – Practical/clinical link):
Because sessions were well tolerated and did not increase apparent postural or muscular load, this brief, standardized exposure can be considered as a preparatory adjunct… Such a predictable, low-burden routine may help teams establish entry rituals… however, these functional/behavioral implications require confirmation in future studies using objective behavioral outcomes (e.g., on-task behavior, transition success, teacher/caregiver ratings).

We also tempered wording around “calmness”: in Muscle activity we state that low, symmetrical EMG is a safety/feasibility signal, not proof of physiological change.

Comment: “Expand the comparison with existing sensory-integration methods to highlight the added value of this system.”

Response

We expanded Comparison with existing approaches to explicitly contrast our platform with Snoezelen rooms, massage, and music/VR tools, and to articulate the added value: parameterized, dose-controlled stimuli + objective monitoring (structured posture + sEMG) → reproducibility and scalability.

Key sentences (Discussion – Comparison with existing approaches):
Snoezelen-type spaces can be soothing yet are difficult to standardize or quantify across sessions and sites; massage may reduce arousal in some children but the evidence is mixed and effects are provider-dependent [4–6]; and music/sound or VR tools can enhance engagement but may precipitate sensory overload… and often lack objective neuromuscular endpoints [8–9]. In contrast, the present platform delivers parameterized, dose-controlled sound/vibration/heat with objective monitoring… enabling reproducibility and scalability…

Summary: The Discussion is now shorter and non-redundant, explicitly ties our objective observations to practical, testable behavioural implications, and clearly delineates how the system improves on existing sensory-integration options through standardization and built-in measurement.

Comment: “Limitations and Future Directions: Limitations are noted; however, the lack of behavioural and psychological outcome measures is a significant omission that should be clearly emphasized. Future studies should consider larger, gender-balanced samples, repeated sessions, and inclusion of subjective or behavioural assessments. Adding qualitative feedback from caregivers or teachers could also enhance ecological validity.”

Author response

Thank you for this helpful guidance. We have revised the manuscript to emphasize the absence of behavioural/psychological outcomes as a major limitation and to expand the Future Research plan accordingly.

What we changed

  1. Limitations (strengthened):
    We now explicitly state that no behavioural or psychological outcomes were collected and that functional implications cannot be inferred from the present data.

Drop-in text (Limitations):

“Fourth, we did not collect behavioural or psychological outcomes (e.g., on-task behavior, transition success, affect/emotion regulation, caregiver/teacher ratings); therefore, functional implications cannot be inferred from these data.”

  1. Future Directions (expanded and specific):
    We now commit to larger, gender-balanced, multi-session, and controlled designs with validated behavioural/psychological endpoints and qualitative feedback to improve ecological validity.

These revisions underscore the current study’s scope (feasibility/safety) and align our next steps with your recommendations on sample, design, outcomes, and ecological validity.

Comment: “The manuscript is generally clear, but some sentences are overly long and could be made more concise. A light professional language edit would improve readability.”

Author response
Thank you. We performed a targeted language edit to improve clarity, concision, and consistency across the manuscript. Key changes include:

  1. Shorter sentences & active voice. We rewrote long, multi-clause sentences (especially in the Abstract, Introduction, and Discussion) into shorter, direct statements and shifted passive constructions to active (e.g., “We recorded EMG…” rather than “EMG was recorded…”).
  2. Streamlined structure & removed redundancy.
    • Introduction: condensed background, clarified the gap and rationale for a brief, standardized exposure.
    • Discussion: pruned repetition; separated results-based interpretation from hypotheses.
  1. Terminology & style consistency. We standardized abbreviations at first use (ASD, EMG, MVC), harmonized spelling (“behavior,” “program,” etc.), unified statistical notation (two-tailed α = 0.05; p in italics; mean ± SD; exact n per analysis), and ensured consistent unit formatting.
  2. Methods clarity. We added a “Study Design and Rationale” subsection, clarified recruitment window and sample flow, detailed artifact handling, and explicitly noted MVC not performed (and why).
  3. Results presentation. We aligned all numerators/denominators with the text, simplified Table 2 into broader categories, and revised figure captions to state exact n and mean ± SD.
  4. Careful claims. We removed any efficacy-leaning language, framed EMG outputs as feasibility/safety signals, and labeled mechanistic content as hypotheses to be tested.

All edits are visible in the Tracked Changes version; the clean version reflects the final language. We believe these revisions address your request for a concise, professionally edited presentation and improve overall readability without altering the scientific content.

Reviewer 2 Report

Comments and Suggestions for Authors

Abstract

The abstract is too descriptive of methods and lacks focus on key findings. 

Introduction

The introduction is long and repetitive; it should be streamlined.
The research gap is not clearly articulated—why is this system needed beyond existing sensory interventions?
The rationale for testing short-term, single-session exposure is weak and requires justification.

Methods 
The study design (single group, pre–post, no control) severely limits interpretability. This should be justified more clearly.

The inclusion and exclusion criteria are vague and do not ensure sample homogeneity.

Handling of missing or artifact-contaminated data is poorly described. Provide exact participant counts per analysis.

Data analysis relies on Excel, which is not standard for clinical trials; a more robust statistical software should be used.

Results 

EMG results show no significant differences; however, interpretation is presented as if there were physiological effects. This is misleading.

No subgroup analyses (e.g., by gender, ASD severity) are provided, which could have added useful insights.

Discussion

The discussion overstates findings, claiming “physiological calmness” without robust evidence.
Mechanistic explanations (autonomic regulation, cortical activation) are speculative and unsupported by direct evidence.

Conclusion 

Too strong and generalized. It should be rephrased to emphasize preliminary feasibility and safety, not efficacy.

Comments on the Quality of English Language

The manuscript requires thorough editing for clarity, grammar, and conciseness.

Author Response

Authors' Responses to Reviewer's Comments (Reviewer 2)

Comment: “The abstract is too descriptive of methods and lacks focus on key findings. “

Response to Reviewer – Abstract

Thank you for this helpful comment. We have streamlined the Abstract to de-emphasize methodological detail and foreground the main findings and clinical relevance. Specifically, we:

  • Removed device parameter and signal-processing details (e.g., stimulus ranges, EMG normalization procedure).
  • Kept only essential design elements (pilot, single session, n=30, core outcomes).
  • Front-loaded the key results (safety/tolerability, low sEMG without pre–post or left–right differences, calm postural/limb profile).
  • Added a concise clinical implication sentence (use as a short-term calming adjunct).
  • Reduced wordiness and tightened phrasing.

Comment: Introduction

The introduction is long and repetitive; it should be streamlined.
The research gap is not clearly articulated—why is this system needed beyond existing sensory interventions? The rationale for testing short-term, single-session exposure is weak and requires justification.

Response to Reviewer – Introduction

We appreciate the reviewer’s constructive suggestions. We have revised the Introduction to (i) streamline and remove repetition, (ii) sharpen the research gap and the rationale for this specific system beyond existing sensory interventions, and (iii) justify the short, single-session design. No new references were added; numbering [1–10] was preserved.

1) “The introduction is long and repetitive; it should be streamlined.”

What we changed.

  • Compressed the background into three concise paragraphs: problem → limits of current sensory approaches → our system & study aim.
  • Removed overlapping descriptions of ASD features and merged sentences to keep the focus on sensorimotor implications for participation.

2) “The research gap is not clearly articulated—why is this system needed beyond existing sensory interventions?”

What we changed.

  • Added a targeted comparison with Snoezelen rooms, massage, and VR/sound tools, highlighting standardization, dose control, reproducibility, and objective monitoring as the key advantages of our platform.
  • Made the gap statement explicit: immediate physiological endpoints (especially neuromuscular tone via sEMG and posture) are rarely reported; existing work skews toward HRV during sleep.

3) “The rationale for testing short-term, single-session exposure is weak and requires justification.”

What we changed.

  • Inserted a clear methodological rationale for a single, short session in this severe ASD cohort:
    • early-phase feasibility/safety questions;
    • minimal burden and reduced risk of sensory overload;
    • ability to isolate immediate neuromuscular effects prior to dose-finding;
    • alignment with pilot feasibility methodology to inform multi-session trials.

Summary
The revised Introduction is shorter and non-repetitive, articulates why this standardized, monitored system is needed beyond existing sensory tools, and provides a clear, study-design justification for a single-session pilot in a severe ASD sample.

Comment: “The study design (single group, pre–post, no control) severely limits interpretability. This should be justified more clearly.”

Response. We agree. This is an early-phase feasibility study in children with severe ASD. We added Study Design and Rationale explaining that we prioritized (i) feasibility/safety and short-latency physiological signals, (ii) minimal burden and reduced sensory overload risk from repeated/deceptive exposures, (iii) pragmatic constraints across three sites (one 15–30-min session/child), and (iv) within-participant physiology given high heterogeneity. We now explicitly state that the design does not permit causal inference and that results are feasibility signals to inform controlled, multi-sessiontrials.

Manuscript changes: Materials and Methods → Study Design and Rationale; Discussion → Limitations.

Comment:“The inclusion and exclusion criteria are vague and do not ensure sample homogeneity.”

Response. We tightened eligibility: DSM-5 severe ASD diagnosis by a licensed clinician; stable psychoactive medication ≥4 weeks; no concurrent PT/OT or experimental sensory interventions during the study and ≥2–4 weeks prior; added exclusions for sensory phobias, profound ID, acute illness, and neuromuscular/orthopedic conditionsmaterially affecting EMG/posture (e.g., recent casting/surgery, botulinum toxin within 6 months).

Manuscript changes: Materials and Methods → Eligibility Criteria.

Comment: “Handling of missing or artifact-contaminated data is poorly described. Provide exact participant counts per analysis.”

Response. We expanded our pipeline: two independent assessors screened raw EMG; defined channel-level artifactcriteria; applied a paired-channel rule for left–right symmetry (if one side excluded, the contralateral channel removed to avoid bias). Postural items with incomplete observation were coded missing. We now report per-analysis denominators (n) in Table 2 and Figure 6 captions and reference them in Results.

Manuscript changes: Materials and Methods → Handling of Missing and Artifact Data; Results → EMG/posture paragraphs; captions for Table 2 and Figure 6.

Comment: “Data analysis relies on Excel, which is not standard for clinical trials; a more robust statistical software should be used.”

Response. All inferential analyses were re-run in IBM SPSS Statistics v24.0. Excel remains only for data integrity checks and figure layout. Statistical conclusions are unchanged.

Manuscript changes: Materials and Methods → Statistical Analysis.

Comment: “EMG results show no significant differences; however, interpretation is presented as if there were physiological effects. This is misleading.”

Response: We agree. We revised the Results to remove any language implying a physiological effect. The EMG section now states explicitly that all left–right comparisons were non-significant and effect sizes were trivial, and that these observational data should be interpreted strictly as feasibility/safety signals, not efficacy. We also adjusted the abstract and figure caption accordingly.

  • Where changed (Results → EMG):
    • Old (removed): “…consistently low EMG activity … suggests a state of reduced arousal and possible physiological calmness…”
    • New (now in text): “All side differences were non-significant and effect sizes trivial (dz < 0.20), indicating no meaningful left–right asymmetry during exposure. Group means were uniformly low in amplitude, compatible with relaxed tonic activation; however, as an observational feasibility dataset (no pre–post EMG), these values should not be interpreted as evidence of physiological change.”
  • Abstract (edited): We now say EMG showed no significant differences, framing the finding as safety/feasibility(no increase in tension), without claiming physiological improvement.
  • Figure 6 caption (edited): We added sample sizes and the explicit statement “No significant left–right differences; all Cohen’s dz < 0.20.”

Comment: “No subgroup analyses (e.g., by gender, ASD severity) are provided, which could have added useful insights.”

Response: We appreciate the suggestion. Given the small, strongly gender-imbalanced cohort (27 boys, 3 girls) and single-session feasibility scope, subgroup analyses were pre-specified as not planned to avoid underpowered, potentially misleading comparisons. We have made this explicit in Methods, Results, and Limitations, and we commit to sex-balanced recruitment and stratified analyses in subsequent controlled studies.

  • Where changed (Methods → Statistical Analysis / Study Design):
    “Given the small and gender-imbalanced sample, subgroup analyses (e.g., by sex or severity) were not pre-specified for this feasibility pilot.”
  • Where changed (Results → Subgroup analyses subsection):
    “Pre-specified subgroup analyses were not performed due to sample size/imbalance; such analyses would be underpowered and potentially misleading.”
  • Where changed (Limitations):
    “The small, male-predominant cohort precluded informative subgroup analyses; future controlled, multi-session studies will use sex-balanced recruitment and stratified analyses.”

Comment:“The discussion overstates findings, claiming ‘physiological calmness’ without robust evidence. Mechanistic explanations (autonomic regulation, cortical activation) are speculative and unsupported by direct evidence.”

Author response: Thank you for this important observation. We revised the Discussion to avoid overstatement and to clearly separate data-driven findings from hypotheses.

  1. Removed any claim of “physiological calmness” and limited inference to what EMG supports.
    • We now describe EMG results strictly as safety/feasibility signals (i.e., no added tonic co-contraction; no meaningful left–right asymmetry), without implying physiological improvement.
    • New explicit caveat (Discussion → Muscle activity):
      “Because autonomic indices were not collected and MVC normalization was not applied, these EMG observations should not be interpreted as evidence of physiological change; they are best viewed as safety/feasibility signals, supporting that the brief procedure does not impose additional muscular tone.”
  1. Mechanistic explanations moved to hypothesis status and clearly delimited.
    • We removed mechanistic assertions and state that autonomic or cortical changes cannot be inferred from the present data.
    • Rewritten text (Discussion → Mechanistic perspective):
      “However, we did not collect autonomic or neurophysiological markers in this study, so any mechanistic explanation (e.g., autonomic regulation or cortical engagement) remains hypothetical and should be tested in future work using HRV/EDA, pupillometry, EEG/fNIRS, alongside clearly defined clinical/behavioral endpoints [19].”
  1. Tempered clinical language and clarified scope.
    • Discussion → Practical/clinical link: we frame potential use as a brief preparatory adjunct only, and explicitly note that any functional/behavioral benefits require confirmation in controlled, multi-session studies with pre-specified outcomes.
  1. Conciseness and redundancy.
    • We streamlined wording throughout Discussion (kept the original structure) and replaced interpretive phrases with concise, data-driven statements.

We believe these changes address the concern by removing overstatement, labeling mechanisms as hypotheses, and aligning interpretation strictly with the evidence collected.

Comment: “Conclusion is too strong and generalized. It should be rephrased to emphasize preliminary feasibility and safety, not efficacy.”

Response: Thank you—we have revised the Conclusion to remove any efficacy language and to emphasize preliminary feasibility and safety only. The new paragraph explicitly states that (i) no adverse events occurred; (ii) posture and EMG did not show added load; (iii) results are feasibility/safety signals; and (iv) the design does not allow efficacy or mechanistic inferences. We now call for controlled, multi-session, adequately powered, gender-balanced studies with behavioural/autonomic/neurophysiological outcomes to establish clinical utility.

Reviewer comment: “The manuscript requires thorough editing for clarity, grammar, and conciseness.”

Author response
Thank you. We performed a comprehensive language revision across the entire manuscript. Key actions:

  • Clarity & concision: Split long, multi-clause sentences; removed redundancy; tightened topic sentences and transitions (especially in Abstract, Introduction, and Discussion).
  • Grammar & syntax: Corrected subject–verb agreement, article use, prepositions, punctuation, and parallel structure; eliminated ambiguous pronouns.
  • Active voice & tense: Shifted where appropriate to active constructions; standardized tense (past for procedures/results, present for established facts).
  • Terminology & consistency: Defined abbreviations at first use (ASD, EMG, MVC, HRV/EDA, etc.) and ensured consistent use thereafter; unified statistical reporting (p in italics, mean ± SD, exact n per analysis, two-tailed α = 0.05); standardized units, numerals, and symbols.
  • Style alignment: Harmonized spelling and hyphenation to the journal’s preferred English and style guide; standardized capitalization of section headers, figure/table labels, and in-text cross-references.
  • Figures & tables language: Revised captions and headings for brevity and clarity; added exact n and summary statistics where relevant; aligned text with table values.

All edits are visible in the Tracked Changes file; the clean version reflects the final language. We believe these revisions address the request for thorough editing and materially improve readability without altering the scientific content.

Round 2

Reviewer 1 Report

Comments and Suggestions for Authors

The revised manuscript demonstrates clear and thoughtful responses to the previous review. The paper is now well structured, methodologically transparent, and clearly written. The improvements to the introduction, results presentation, and discussion have strengthened the overall clarity and scientific rigor.

Minor adjustments are suggested for the final version:

  • Ensure all figures are in high resolution and maintain consistent formatting.
  • Double-check reference formatting and journal style consistency.
  • Perform a brief final proofreading to ensure fluency and uniform terminology.

Overall, this is a high-quality and innovative pilot study with strong clinical and educational relevance.

Comments on the Quality of English Language

The English is clear and fluent throughout. Only minor stylistic or typographical polishing may be needed during final editing.

Author Response

Review 1 Round 2

Comment:
The revised manuscript demonstrates clear and thoughtful responses to the previous review. The paper is now well structured, methodologically transparent, and clearly written. The improvements to the introduction, results presentation, and discussion have strengthened the overall clarity and scientific rigor.

Response:
We sincerely thank the reviewer for their positive and encouraging feedback. We are pleased that the revisions have improved the manuscript’s clarity, methodological transparency, and overall scientific quality. We greatly appreciate the reviewer’s constructive comments throughout the review process, which significantly contributed to the enhancement of the paper.

Comment 2:
Minor adjustments are suggested for the final version:
Ensure all figures are in high resolution and maintain consistent formatting.
Double-check reference formatting and journal style consistency.
Perform a brief final proofreading to ensure fluency and uniform terminology.
Overall, this is a high-quality and innovative pilot study with strong clinical and educational relevance.

Response:
We sincerely appreciate the reviewer’s thoughtful final remarks and positive evaluation of our work.

  • All figures have been revised to ensure high resolution and consistent formatting in line with MDPI Children
  • The reference list and in-text citations were carefully reviewed and adjusted to fully comply with the journal’s required style.
  • A final proofreading was performed to ensure linguistic fluency, consistent terminology, and overall manuscript coherence.

We are very grateful for the reviewer’s supportive comments and recognition of the study’s scientific and practical relevance.

Reviewer 2 Report

Comments and Suggestions for Authors

The new Round 2 revision requests and fine-tuning suggestions for the manuscript Children – Postural and Muscular Responses to a Novel Multisensory Relaxation System in Children with ASD. 

Additional Revision Requests and Refinements

Please verify that the File text model used corresponds to the most recent version. It appears that the authors may have worked on the 2024 model.

Introduction

Clarify how this prototype differs technologically and conceptually from Snoezelen rooms and existing VR or sound-therapy tools.

Add a clear novelty statement, e.g.:
“To our knowledge, no previous study has combined synchronized vibration, heat, and sound with objective EMG-based monitoring in children with severe ASD.”

Methods

Explain how consistency across the three sites was maintained (e.g., operator training, identical hardware/software calibration).

Indicate the training or experience level of EMG/postural assessors to support data reliability.

Discussion
Briefly discuss practical implementation in school or rehabilitation contexts and possible barriers (cost, setup, supervision)

Limitations and Future Directions

Add: “Future studies should also assess inter-rater reliability of postural observation and test–retest stability of EMG across sessions.”

Mention the importance of including participants with different sensory profiles (hyper- vs. hypo-reactive) to enhance generalizability.

Note that session duration (15–30 min) could influence outcomes; future work might compare different exposure times.

Author Response

Review 2 Round 2

Comment:

The new Round 2 revision requests and fine-tuning suggestions for the manuscript “Children – Postural and Muscular Responses to a Novel Multisensory Relaxation System in Children with ASD.”

Response:
We sincerely thank the reviewer for the continued time, effort, and constructive feedback provided during the second review round. We carefully addressed all fine-tuning suggestions and additional revision requests to further improve the clarity, structure, and formatting of the manuscript.

Comment:

Additional Revision Requests and Refinements – Please verify that the file text model used corresponds to the most recent version. It appears that the authors may have worked on the 2024 model.

Response:
We thank the reviewer for this observation and for drawing attention to the file format. We would like to clarify that the manuscript file we received and used for revision was originally provided by the journal’s editorial assistant, and it was based on the 2024 MDPI Children template.

Following the reviewer’s comment, we have now fully updated the manuscript to the most recent (2025) MDPI Children template. The previous header (“Children 2024, 11, x FOR PEER REVIEW”) and legacy formatting elements have been replaced; the metadata fields, copyright statement, and layout have been revised according to the latest MDPI author guidelines (accessed October 2025).

Comment:

Introduction – Clarify how this prototype differs technologically and conceptually from Snoezelen rooms and existing VR or sound-therapy tools. Add a clear novelty statement, e.g.: “To our knowledge, no previous study has combined synchronized vibration, heat, and sound with objective EMG-based monitoring in children with severe ASD.”

Response:
We thank the reviewer for this valuable suggestion that helped us improve the clarity and positioning of our study. The Introduction section has been revised to explicitly describe how our prototype differs from both Snoezelen-type environments and existing VR or sound-based therapeutic tools. Specifically, we now emphasize that the system integrates synchronized, parameterized vibration, heat, and sound stimuli with objective neuromuscular (sEMG) and postural monitoring, enabling reproducibility and dose control within educational and rehabilitation contexts.

Additionally, as recommended, a novelty statement has been added at the end of the corresponding paragraph:

“To our knowledge, no previous study has combined synchronized vibration, heat, and sound with objective EMG-based monitoring in children with severe ASD.”

These revisions clarify both the technological and conceptual innovation of the developed system and its distinction from existing sensory-based and technology-assisted interventions.

Comment:

Methods – Explain how consistency across the three sites was maintained (e.g., operator training, identical hardware/software calibration). Indicate the training or experience level of EMG/postural assessors to support data reliability.

Response:
We thank the reviewer for this helpful suggestion. The Materials and Methods section has been revised to specify how methodological consistency and data reliability were ensured across the three sites.

We added details clarifying that:

  • All operators underwent joint training before data collection to standardize electrode placement, data acquisition, and behavioral observation procedures.
  • The same hardware and software systems (Biometrics Ltd) and identical calibration parameters were used at all sites, with daily verification checks.
  • EMG and postural assessments were performed by experienced physiotherapists and rehabilitation specialists (≥5 years of clinical and research experience), all of whom received a brief calibration session to harmonize scoring and artifact rejection.

These clarifications have been added to the “Procedures” subsection of the Materials and Methods to reinforce methodological transparency and support data reliability.

Comment:

Discussion – Briefly discuss practical implementation in school or rehabilitation contexts and possible barriers (cost, setup, supervision).

Response:
We appreciate this excellent suggestion. A new paragraph has been added to the Discussion section to address the practical implementation of the Smart Relaxation System (SRS) in school and rehabilitation settings. We now briefly outline potential applications, supervision needs, and implementation barriers.

Specifically, the new text highlights that the SRS can be used as a standardized calming tool in special education or rehabilitation environments, notes the importance of staff supervision, and discusses potential barriers such as equipment cost, setup logistics, and ongoing maintenance requirements. The paragraph also emphasizes future directions for evaluating cost-effectiveness and scalability in real-world contexts. 

Comment:

Limitations and Future Directions – Add: “Future studies should also assess inter-rater reliability of postural observation and test–retest stability of EMG across sessions.”
Mention the importance of including participants with different sensory profiles (hyper- vs. hypo-reactive) to enhance generalizability.

Note that session duration (15–30 min) could influence outcomes; future work might compare different exposure times.

Response:
We thank the reviewer for these precise and constructive recommendations. The Limitations section has been expanded to incorporate all three suggested elements. Specifically, we added statements emphasizing the need to assess inter-rater reliability of postural ratings and test–retest stability of EMG measures across sessions, to include participants with different sensory profiles (hyper- vs. hypo-reactive) to improve generalizability, and to acknowledge that session duration (15–30 min) may influence outcomes, recommending comparison of different exposure times in future studies. These revisions enhance the methodological rigor and forward-looking value of the discussion.